# GLPS: A Geohash-Based Location Privacy Protection Scheme

**DOI:** 10.3390/e25121569

**Published:** 2023-11-21

**Authors:** Bin Liu, Chunyong Zhang, Liangwei Yao, Yang Xin

**Affiliations:** 1National Engineering Research Center for Disaster Backup and Recovery, Information Security Center, School of Cyberspace Security, Beijing University of Posts and Telecommunications, Beijing 100876, China; ahstuliubin@bupt.edu.cn (B.L.); chertish_xxd@bupt.edu.cn (C.Z.); yaoliangwei_2006@163.com (L.Y.); 2College of Information and Network Engineering, Anhui Science and Technology University, Bengbu 233030, China

**Keywords:** location-based services, location privacy protection, Geohash code, location semantics

## Abstract

With the development of mobile applications, location-based services (LBSs) have been incorporated into people’s daily lives and created huge commercial revenues. However, when using these services, people also face the risk of personal privacy breaches due to the release of location and query content. Many existing location privacy protection schemes with centralized architectures assume that anonymous servers are secure and trustworthy. This assumption is difficult to guarantee in real applications. To solve the problem of relying on the security and trustworthiness of anonymous servers, we propose a Geohash-based location privacy protection scheme for snapshot queries. It is named GLPS. On the user side, GLPS uses Geohash encoding technology to convert the user’s location coordinates into a string code representing a rectangular geographic area. GLPS uses the code as the privacy location to send check-ins and queries to the anonymous server and to avoid the anonymous server gaining the user’s exact location. On the anonymous server side, the scheme takes advantage of Geohash codes’ geospatial gridding capabilities and GL-Tree’s effective location retrieval performance to generate a *k*-anonymous query set based on user-defined minimum and maximum hidden cells, making it harder for adversaries to pinpoint the user’s location. We experimentally tested the performance of GLPS and compared it with three schemes: Casper, GCasper, and DLS. The experimental results and analyses demonstrate that GLPS has a good performance and privacy protection capability, which resolves the reliance on the security and trustworthiness of anonymous servers. It also resists attacks involving background knowledge, regional centers, homogenization, distribution density, and identity association.

## 1. Introduction

With the development of mobile applications, location-based services (LBSs) have been widely used. Snapshot query and trajectory query are the two main service categories of LBSs. Among them, the snapshot query service is widely used and is the basis of the trajectory query service. It is the most fundamental supporting technology in LBS.

LBSs require users to submit their personal real-time location and query content. If a malicious adversary obtains this location and content information, it can further mine other private information about the user. To protect users’ privacy, researchers have conducted numerous studies.

### 1.1. Typical Location Privacy Protection Technologies

Researchers usually focus on security within the life cycle of application services. For example, for the security of software deployment, TPM (Trusted Platform Module) computing technology is used to ensure the integrity of program code [1]. For network attacks and system intrusion, attack process identification and prevention techniques are studied. From the perspective of system availability and disaster recovery, research on disaster recovery and backup technologies has been conducted to ensure data security and business sustainability [2,3]. In contrast to the above research, research on location privacy protection technology assumes that the user’s location and query are not accurately recognized as much as possible when the system has been intruded by the attacker, which in turn ensures the user’s privacy security. Therefore, location privacy protection schemes usually adopt the uncertainty idea. They achieve privacy protection by reducing the probability that an adversary is aware of the exact location of the user and the query content. Typical location privacy protection technologies are *k*-anonymity, dummy, spatial cloaking, spatial transformation, release suppression, PIR (privacy information retrieval), encryption, differential privacy, and their combination.

The *k*-anonymity algorithm [4,5,6,7] has the property of probabilistic indistinguishability. It is widely used in location privacy protection schemes. It mixes the user’s location with other k−1 assisted locations to construct a *k*-anonymous location set. Thus, the attacker cannot accurately associate the user’s identity and exact location from the *k*-anonymous set. Theoretically, a well-designed algorithm can ensure that the probability of an attacker determining the exact location of a user dose not exceed $\frac{1}{k}$. The larger the value of *k*, the better the location-hiding effect.

Randomized dummy locations and false queries are also commonly used [8,9,10,11,12,13]. These schemes require that randomly generated dummy locations and queries should be not only logically sound but also probabilistically indistinguishable from the real user. These methods also introduce additional time delays, computation, and communication.

Spatial cloaking [14,15,16] hides the user’s location in a spatial location region, making it impossible for an attacker to accurately identify the user’s exact location from the cloaking region. The area of the cloaking region positively correlates with the privacy-preserving capability and negatively correlates with the quality of service. Therefore, users need a balance between privacy protection and service quality.

The spatial transformation uses space-filling curves to convert the user’s exact location coordinates into a one-dimensional code (such as Hilbert [17,18,19] and Geohash [20]). These methods are widely used in PoI (Points of Interest) and NN (Nearest Neighbors) queries. Since the code corresponds to the user’s location interval, the essence of the idea is equivalent to the spatial cloaking scheme.

Mix-zone [21,22] schemes use a release suppression technique. These schemes use a mix-zone containing multiple users, in which users are stopped from sending query requests and their identities are randomly replaced, thus cutting off the association between user trajectories and identities. However, these schemes restrict the use of location services in the mix-zone, which affects the user’s service experience.

Location substitution uses other locations near the user (e.g., landmarks [23,24]) to send the query instead of the user’s location. This method leads to inaccurate query results.

PIR [25,26] and homomorphic encryption [27,28,29] techniques use cryptography to send queries to the server in ciphertext form and obtain query results based on ciphertext computation. This method provides a more secure information protection capability, but the complex cryptographic algorithm increases the consumption of storage and computational resources and reduces the performance of the retrieval response.

The differential privacy scheme has a well-developed mathematical model and is effective against background knowledge attacks [30,31,32,33]. However, the noise introduced corrupts the original data and reduces the accuracy and usability of the data.

Existing location privacy protection techniques inevitably have certain limitations. Researchers need continuous research on location privacy protection in terms of architecture, methodology, and evaluation metrics to find solutions that balance service quality, query efficiency, and protection capabilities.

The technical solution used in this paper is a combination of spatial hiding and *k*-anonymity techniques, and the following discussion focuses on some of the studies that are more relevant to the research in this paper.

### 1.2. Research on Related Location Privacy Protection Solutions

Gruteser et al. [4] pioneered *k*-anonymity location privacy-preserving techniques. They chose k−1 locations to hide in a rectangular or circular cloaking area with the user’s location and sent queries to LSPs with the cloaking area instead of the user’s exact location. This study pioneered a technological direction in location privacy protection research but failed to provide personalized user privacy requirements.

To meet users’ personalized privacy needs, CliqueCloak [5] used the minimum level of anonymity as the user’s privacy requirement and the maximum temporal and spatial tolerance as the service quality criteria. Casper [14] used the minimum anonymity level and the minimum acceptable area of the spatial region as the user’s privacy requirements. However, these algorithms ignored the effect of population density on the size of the cloaking area and service quality. In densely populated areas, there is less hiding space and a high risk of user location leakage. In sparsely populated areas, there is more space to hide, and the quality of service declines.

To reduce the computational load on the user side, many researchers have adopted centralized architectures and introduced anonymous servers (ASs) to construct location privacy protection algorithms [15,16]. QuadTree has been used to store and efficiently retrieve massive location areas on ASs. QuadTree iteratively and uniformly splits a geographic region into many small subregions, recording the list of users or the density of users within the subregions. An AS retrieves the QuadTree, selects the user and other users in the neighboring regions, and jointly constructs the *k*-anonymous set that satisfies the user’s privacy requirements. These schemes are built on the assumption of absolute security and trustworthiness of the AS. However, that assumption is not applicable in reality.

The Aman algorithm [34] removed the AS and introduced a Density Cloud Server. The Density Cloud Server uses a QuadTree to construct the spatial distribution information of the users and provide query services for the users. The scheme assumed that users were uniformly distributed in the location space, and based on this assumption, it predicted the hierarchy of location retrieval to improve the retrieval speed. However, the uniform spatial distribution of users does not correspond to reality, and the scheme increased the communication cost between users and the density cloud.

The dummy position solution is now also widely used. Lu et al. [8] proposed CirDummy and GridDummy schemes, which use circular and square geographic regions to hide user locations. They divided the geographic area into several sub-regions of equal size. The user’s location and other random dummy locations are uniformly distributed in these sub-regions to increase the difficulty of guessing. However, the scheme only focused on the uniformity of the spatial distribution of locations in the anonymous set but ignored the semantic rationality of the dummy locations. Niu et al. [9] proposed the DLS and EDLS schemes, based on historical query records, to select cells with similar query probabilities to the user’s cell as the fake locations among the divided spatial cells to avoid attackers from filtering the false cells based on the query probability distribution. However, the historical queries do not accurately reflect the real-time characteristics of the location, and these algorithms still greatly depend on the security and trustworthiness of the AS.

### 1.3. Research on Geohash in Location Services and Privacy Protection

Geohash [35] is a geolocation encoding technique. It can convert precise two-dimensional geographic coordinates (latitude, longitude) into one-dimensional string identifiers (Geohash codes) by iteratively dividing the Earth’s surface into several grid regions of controllable size and using region coding. As a result of its features of simple coding, efficient retrieval, spatial generalization, and personalized code length definition, Geohash encoding has been applied in the research of user localization, area monitoring, proximity query, spatial object recognition, and location privacy protection.

Using the Geohash code gridded area capability, Huang et al. [36] counted and analyzed COVID-19 epidemic-related data; Zhang et al. [37] studied the trajectory of a patient with an infectious disease for trajectory identification; Ding et al. [38] studied automatic maize precision planter’s partitioning control; Liu et al. [39] designed a hierarchical clustering algorithm for smart-city applications; and Miao et al. [40] designed a privacy-preserving spatial range query (PSRQ) scheme.

Using the locations of Geohash codes with the same prefix, which are geographical neighbors, Wu et al. [41] proposed a geospatial-aware deep trajectory representation learning method to study the similarity of trajectories. Verma et al. [42] constructed a Geohash-labeled graph for cab movement detection and trajectory prediction, and Guo et al. [43] investigated the spatial range query, nearest neighbor query, and spatial object size query.

For location privacy protection, researchers have devised schemes using Geohash codes. Yin et al. [19] designed a location privacy protection scheme using a stand-alone system architecture. They gridded the user’s extended interval and constructed a *k*-anonymous location set by selecting grids with similar access probabilities to the user’s grid based on the historical query probability. Wei [44] encoded the user’s location coordinates, achieved spatial hiding of the exact location, and found PoIs in the user’s grid and its eight surrounding grids. Gao et al. [45] encoded trajectories covering different grids and designed Geohash-Tree query trees to speed up the efficiency of querying. Liu et al. [46] proposed a location privacy protection scheme combining Geohash and pseudo-random sequence techniques. Xing et al. [47] designed the G-Casper scheme, which uses the user’s Geohash-encoded location grid and its surrounding eight grids as candidate grids, and selected one or more grids to construct the hidden intervals through the processing the random grid numbering, checking the number of users, and pruning. Zhang et al. [48] incorporated the height coordinates of user locations and designed a *k*-anonymous location privacy protection strategy based on the Alt-Geohash encode. All these schemes assume that the AS is absolutely secure and trustworthy, but in reality, no such AS exists, because the AS is at the central key position of the architecture and thus it is more vulnerable to attack and to becoming a performance bottleneck for the system.

To address the problem of over-reliance on the security and trustworthiness of ASs, Chen et al. [49] proposed a two-server architecture with service splitting. They divided the Geohash code into two parts: a prefix and suffix. Based on the Geohash prefix submitted by the user, the social network server filters out users with the same prefix to form a set of candidate neighbors. The third-party server obtains the encrypted suffix from the user side, completes the decryption and relative location coordinate approximation replacement, and selects the final result from the candidate neighbor set. However, this scheme increases the computational burden on the system.

Some methods have included server clustering to improve query efficiency and permit concurrent requests. Ni et al. [50] proposed a scheme using a distributed memory–object–cache–server cluster. They used a trusted agent to transform the user’s location coordinates into Geohash codes and store PoI information in the form of <Geohash, PoI> mapping pairs on different nodes of the cache server cluster for parallel querying. However, they failed to account for the possibility of surrounding location encoding mutations in Geohash, which could lead to the assignment of PoIs at neighboring locations to different nodes in the cluster, producing query responses that are both insufficient and erroneous. Zhou et al. [51] used a load-balancing server and a cluster of PoI query servers to achieve concurrent queries. Queries from users using Geohash codes are sent to the load-balancing server. The load-balancing server redirects them to a specific PoI server in the cluster based on the PoI type of the query. The PoI server constructs a trie tree, storing the Geohash codes of PoI locations in advance. After receiving the query, the server performs character matching retrieval in the trie tree based on the Geohash code of the user’s location to find the PoI location of the nearest neighboring location. This method’s disadvantage is that each server in the cluster only stores and processes the associated PoI query, which will result in an unequal server load due to the distributions of the total amount of various PoI data types and query volume in both space and time.

In conclusion, the location service and privacy protection schemes based on Geohash encoding technology can realize location spatial hiding and efficient retrieval using Geohash code strings instead of location coordinates. Schemes that deploy the Geohash encoding process at the user’s end enable regional privacy of the location so that the query recipient does not know the exact location of the user, but these schemes have a reduced service accuracy. Schemes that deploy the Geohash encoding process at the anonymous server require the user to send the exact location to the anonymous server, so the anonymous server needs to be secure and trustworthy. However, the user can get more accurate query results.

### 1.4. Contributions of This Article

To address the problems of a centralized architecture, the motivation behind this paper is to combine *k*-anonymity, spatial cloaking, and Geohash encoding techniques to design a location privacy protection scheme for snapshot queries (named GLPS). This scheme can avoid the strong dependence on the security and trustworthiness of ASs and achieve resistance to various types of attacks (including identity association attacks, background knowledge attacks, regional center attacks, homogenization attacks, and distribution density attacks).

Our main contributions are as follows:At the user’s terminal, we exploit the spatial hiding ability of GeoHash codes to hide the exact location coordinates of the user in a spatial region. The user uses the location’s Geohash code as a privacy location to check in to an AS, allowing them to initiate location service queries or participate in an anonymous query set without revealing their exact location.On the AS, we use GL-Tree to achieve the efficient storage and retrieval of user location intervals to provide data support. We design a new algorithm for *k*-anonymous query set generation. The algorithm further extends the user’s cloaking area based on the user’s minimum and maximum hidden cells to meet the user’s personalized location privacy requirements.We define the *cell hotness feature* and *cell behavioral feature* to characterize the user preference and query behavior of the cell, and based on these, we further define the *cell semantic features* and construct a *cell semantic feature matrix*. This matrix reflects the semantic features of a cell in *m* consecutive observation periods.We define the cell similarity calculation method for quantifying the degree of semantic feature similarity between two cells.We experimentally tested the scheme on the public dataset GeoLife and conducted comparative experiments and analytical discussions with Casper, *G-Casper*, and DLS.

The rest of the paper is organized as follows. Section 2 presents the relevant preparatory knowledge. Section 3 discusses the system architecture, adversary assumptions, formal description, algorithmic ideas, pseudo-code, and algorithmic analysis of our scheme. In Section 4, we experimentally test the GLPS scheme on the GeoLife public dataset and analyze it and compare it with Casper, *G-Casper*, and DLS schemes to verify the effectiveness and security of GLPS. Section 5 summarizes the advantages of GLPS and discusses future work.

## 2. Preliminaries

### 2.1. Typical Attacks

1.Identity association attack

Although the *k*-anonymous scheme reduces the probability that an attacker will determine the precise location of a user, when an attacker obtains enough historical query records through some means, the attacker can analyze the historical records to find the location features and thus establish a correlation between the user’s identity and corresponding query location and query content.

2.Background knowledge attack

The boom of big data, mobile terminals, APPs, the IoT, and information service platforms has given adversaries more opportunities and methods to obtain specific background knowledge (e.g., demographic data, city traffic network, PoI geographic distribution, municipal planning). This knowledge helps attackers to exclude false locations and queries. For instance, an attacker can rule out randomly produced fake locations in geographically uninhabited or inaccessible regions. An attacker can find bogus positions and false trajectories using a road network distribution map if they do not fit the topology of the road network.

3.Regional center attack

Some algorithms expand the search by centering on the user’s location to find associate users and to construct a *k*-anonymous set together with the user. This results in a high probability that the user’s location is at or near the center of the cloaking area. An attacker can utilize this feature to easily infer the user’s exact location or significantly reduce the range of probable locations.

4.Homogenization attack

In reality, the behavioral characteristics of people in the same geographical area often show some similarities. People exhibit particular behavioral characteristics in densely populated areas, such as factories, schools, office buildings, and hospitals. For example, in hospital areas, people are more likely to engage in activities related to medicine, treatment, and health. Therefore, if the locations in the anonymous set are relatively centralized or in the same geographical area, there is a greater likelihood that the queries in the anonymous query set will exhibit homogenization. This homogeneity makes it possible for an attacker to easily infer a user’s private data through the geographical coverage of the cloaked area and the content of the queries, without knowing the user’s exact location and identity.

5.Distribution density attack

In reality, the geographical distribution of the population is very uneven. This heterogeneity is usually strongly associated with geography, location function, time, and other factors. Population densities in the same area vary at different times. For example, plains are more densely populated than barren areas. Urban areas are more densely populated than rural areas. Restaurants and food areas are densely populated during dining hours. Schools are densely populated during teaching times. Population distribution density characteristics not only affect the spatial size of the cloaking area but also may be exploited by attackers to help them identify fake locations.

### 2.2. Geohash Encoding

Geohash is a geolocation-encoding technique that enables efficient retrieval. It can convert two-dimensional geographic location coordinates into an ASCII string identifier (Geohash code) of a specific length. Efficient location retrieval and nearest neighbor retrieval can be achieved by string matching. Each Geohash code corresponds to a rectangular geographic location interval, and the area of the interval is determined by the length of the code. The specific principle and pseudo-code of Geohash encoding can be found in [35,52]. According to the Geohash encoding, we can obtain the following basic properties:A location’s coordinate can be converted into a unique Geohash code. However, a Geohash code represents a rectangular geographic region. All locations in the region have the same Geohash code.The longer the length of a Geohash code, the smaller the area of the rectangular geographic region it represents.Each additional character in the Geohash code string is equivalent to dividing the region corresponding to the original Geohash code into 32 equal-sized grid sub-regions, and these 32 grid sub-regions are numbered according to the Z-shaped space-filling curve.Two Geohash codes with the same string prefix and their corresponding geographic regions belong to the same larger region, and the Geohash code of that region is this co-prefix string.Regions with the same Geohash code prefix are also geographically adjacent, and the longer the same prefix, the closer the geographic location.

Therefore, by matching Geohash codes with the same prefix string, one can find the location of spatial nearest neighbors, avoiding the high consumption introduced by the Euclidean distance calculation. Releasing the user’s location using GeoHash codes of different lengths hides the user’s exact location within rectangular geographic regions of different sizes.

### 2.3. GL-Tree

GL-Tree is a hierarchical tree data structure that we designed for the storage, management, and efficient retrieval of large areas in our previous research [52]. It reflects the correspondence between Geohash codes and the grid division of rectangular geographic regions. For each level of vertical expansion of the hierarchy, the length of Geohash code is increased by one. A region of Geohash code length *L* corresponds to a datum in the *L*-th layer of the GL-Tree. Each level of the GL-Tree consists of multiple L-Trees. Each L-Tree corresponds to a Geohash code rectangular geographic region and 32 equal-sized grid sub-region divisions of this region. Each L-Tree has 32 data items, and each of them corresponds to one of these 32 grid sub-regions. The keyword in a datum is one of the Base32 encoded characters. Efficient storage and retrieval of locations can be achieved by following the GL-Tree in a top-down manner. For a description of the structure, characteristics, physical meaning, retrieval and maintenance process, and experimental comparison of GL-Tree, please read [52].

To make the GL-Tree suitable for the scheme described in this paper, we have adapted the data items of the GL-Tree so that each data item can store the check-ins and history queries in its corresponding rectangular grid region (also named a cell). Figure 1 shows the modified structure. We set a binomial tree cs and a one-way list hl in each data item. The cs stores online user check-ins, and the hl stores historical queries. The cs cannot store duplicate data, but the hl can. We set a user to be recorded only for the latest check-in within a cell. A user may send multiple queries in a cell, each of which can be stored. This setup is in line with the actual LBS application scenario.

## 3. The Proposed Scheme

### 3.1. System Architecture

To reduce the complexity of the system structure and make the designed scheme applicable to lightweight user terminals, we use a centralized architecture to build the location privacy protection scheme. As shown in Figure 2, the system architecture includes the user terminal device (user), check-in/anonymous server (AS), and location service provider’s servers (LSPs). The basic working process is briefly described as follows, and the details will be elaborated in the algorithm section.

The AS reads historical query records from the database and initializes the GL-Tree.The user uses a terminal with a positioning function to obtain real-time geographic location coordinates. Then, according to the personal privacy space requirements, they select a suitable code length and convert their location coordinates into Geohash code. The user sends a location check-in to the AS with this Geohash code as the privacy location *g* instead of the exact location coordinates.The AS receives a check-in and inserts the check-in information into the GL-Tree or updates the check-in record.A user sends a query to the AS. The query contains the user’s identity, privacy location, time, query content, and personalized privacy requirements.The AS uses the algorithm proposed in this paper to construct a *k*-anonymous query set based on users’ individual privacy requirements.The AS sends the *k*-anonymous query set to the LSPs.The LSPs receive the *k*-anonymous query set, process each query in the set, generate the corresponding results set, and send it back to the AS.The AS filters the exact query result of the user from the results set and returns it to the user.

### 3.2. Adversary Assumptions

To reduce the high dependence of the scheme on the security and trustworthiness of any entity in the system, we make the following adversary assumptions.

We assume that the deployment process of the application service is secure and that the communication network does not leak information during operation. The deployment process and the security measures of the communication network are beyond the scope of this paper.We assume that users are semi-trustworthy (honest but curious). Users will faithfully execute the algorithmic process of the scheme, but users may probe other users’ information out of curiosity.We assume that LSPs are a potential adversary. The LSPs can faithfully fulfill the agreement and fulfill the duties of the location query service, but their security is not trustworthy because of the possibility of being compromised by an attacker. In addition, location service providers may also maliciously mine users’ private information or directly disclose users’ query records to unauthorized third parties for commercial gain.We consider that the AS is a semi-trustworthy (honest) adversary. It faithfully executes the anonymous query set construction method, does not actively leak user queries or check-ins, and does not actively collude with LSPs. However, there is a chance that attackers will gain access to the AS. Therefore, it is necessary to restrict how precisely it can determine the user’s location.We assume that the adversary has access to the algorithms used by the scheme and has access to historical queries and geographical background knowledge. The background knowledge is limited to the geographical environment, population distribution, and road network information. The adversary attempts to use this information to deduce the precise location and content of the user’s query from the anonymous query set, thereby mining the user’s privacy.

### 3.3. Formal Description

In this section, we provide formal descriptions of several concepts used in the scheme.

**Definition 1** (User’s Location)**.** 
*The scheme uses two ways to describe the user’s location: precise location and privacy location. The former is a two-dimensional coordinate point in geospace, denoted by the binary loc=(x,y), where x is the latitude and y is the longitude. The latter is a Geohash code of specified length corresponding to the exact location, denoted as $g=Geohash(loc,g_{len})$. The g corresponds to a rectangular area of the user’s location in geospace. Its size is related to the specified Geohash code length $g_{len}$. The g limits the accuracy with which the AS can know the user’s exact location. The user adjusts the size of the minimum hidden area by setting $g_{len}$. The larger the $g_{len}$, the smaller the area, and vice versa. g values of different lengths indicate that users use different sizes of rectangular areas as hidden regions.*


**Definition 2** (Cell)**.** 
*A Geohash code corresponds to a rectangular area of geospace, which we call a cell in this paper.*


**Definition 3** (Check-in)**.** 
*Users check in to the AS before using the location-based query service or participating in anonymous set construction. A check-in is defined as a triple consisting of uid (the user identifier), g (the user’s privacy location), and t (the check-in time) together, denoted as checkin=(uid,g,t).*


**Definition 4** (Query)**.** 
*A query submitted by a user for a location-based service. A query is defined as a five-tuple consisting of uid (the user identifier), g (the user’s privacy location), t (the time), c (the content), and p (a personalized privacy protection metric), denoted as Q=(uid,g,t,c,p).*


**Definition 5** (Privacy-Preserving Metric)**.** 
*The privacy-preserving metric reflects the user’s personalized privacy requirements for a query. We define the privacy-preserving metric as a triple p. p consists of k (anonymity), $g_{min}$ (the length of the Geohash code corresponding to the maximum hidden region), and $g_{max}$ (the length of the Geohash code corresponding to the minimum hidden region), denoted as $p=(k,g_{min},g_{max})$. Note that the longer the Geohash code, the smaller the hidden region area. Generally, $g_{max}\leq g_{len}$.*


**Definition 6** (Anonymous Query)**.** 
*A query in an anonymous query set is denoted as $Q^{\prime }$. The $Q^{\prime }$ is a quadruple consisting of pid (the user’s pseudonym), g (the privacy location), t (the query time), and c (the query content). $Q^{\prime }=\left(pid,g,t,c\right)$.*


**Definition 7** (Anonymous Query Set)**.** 
*In a query service, based on a user’s query, the AS generates a set of queries that satisfy the user’s privacy needs, denoted as $Qs=\{Q_{1}^{\prime },Q_{2}^{\prime },\ldots ,Q_{k}^{\prime }\}$.*


**Definition 8** (Anonymous Range)**.** 
*The algorithm will construct a hidden region for each query, denoted by AR. It consists of several cells. In there, the AS can select other k−1 users to build anonymous groups with the user to satisfy p requirements for k-anonymity.*


**Definition 9** (Cell hotness Feature)**.** 
*As a result of the differences in geographic location, service functions, and service hours, the distribution of users in cells usually shows the characteristics of spatial and temporal distribution unevenness. This reflects the performance of cell features and the reflection of users’ preference for cells. We use the cell hotness feature to measure the degree of user preference for a specific cell at different periods and represent it as a time series vector CH, e.g., $CH_{i}=\left(CH_{i}^{1},CH_{i}^{2},\ldots ,CH_{i}^{j},\ldots ,CH_{i}^{m}\right)$ denotes the cell hotness feature of the i-th cell in m consecutive periods.*

(1)
$$\begin{array}{c} CH_{i}^{j}=\frac{\left|UC_{i}^{j}\right|}{\sum_{i=1}^{n}\left|UC_{i}^{j}\right|} \end{array}$$


(2)
$$\begin{array}{c} \sum_{i=1}^{n}CH_{i}^{j}=1 \end{array}$$

*where i denotes the cell number, $i\in \left[1,n\right]$, j denotes the time period, $j\in \left[1,m\right]$, n denotes the total number of cells in the user’s maximum hidden area, and $\left|UC_{i}^{j}\right|$ denotes the number of checked-in users of the i-th cell in the j-th time period.*


**Definition 10** (Cell Behavioral Features)**.** 
*Usually, users exhibit various behavioral characteristics in different cells. Even if a user is in the same cell, their behavioral characteristics differ in different periods. The cell behavioral features are quantitative measures of the user’s query behavior in a particular cell at different periods. We use two time series vectors, request feature CQ and content feature CT, to quantify cell behavioral features. $CQ_{i}=\left(CQ_{i}^{1},CQ_{i}^{2},\ldots ,CQ_{i}^{j},\ldots ,CQ_{i}^{m}\right)$ denotes the request feature vector of the i-th cell in consecutive m periods. $CT_{i}=\left(CT_{i}^{1},CT_{i}^{2},\ldots ,CT_{i}^{j},\ldots ,CT_{i}^{m}\right)$ indicates the content feature vector of the i-th cell in consecutive m periods. The symbol $CQ_{i}^{j}$ indicates the proportion of users sending queries for the i-th cell in the j-th period. Symbol $CT_{i}^{j}$ indicates the proportion of the number of query content types in the i-th cell to the number of all query content types in all n cells in the j-th period. The smaller the value, the more the type of user query content converges in the cell.*

(3)
$$\begin{array}{c} CQ_{i}^{j}=\frac{\left|UQ_{i}^{j}\right|}{\left|UC_{i}^{j}\right|} \end{array}$$


(4)
$$\begin{array}{c} CT_{i}^{j}=\frac{\left|UT_{i}^{j}\right|}{\left|{⋃}_{i=1}^{n}UT_{i}^{j}\right|} \end{array}$$

*where $\left|UQ_{i}^{j}\right|$ and $\left|UT_{i}^{j}\right|$ denote the number of users sending query requests and the number of query types in the j-th period for the i-th cell, respectively.*


**Definition 11** (Cell Semantic Features)**.** 
*We use cell hotness and behavior features to portray cell semantic features and form a quantization matrix, $M_{i}=\left[CH_{i},CQ_{i},CT_{i}\right]$. $M_{i}$ is an m×3 numerical matrix. It reflects the regional semantic features of the cell in m consecutive periods.*


**Definition 12** (Cell Semantics Similarity)**.** 
*This is used to measure the degree of similarity between the semantic feature matrices of two cells. We use the mean value of the similarity of the regional semantic vectors of the two cells in each period as the cell semantic similarity. D denotes the Euclidean distance between two vectors.*

(5)
$$\begin{array}{c} sim\left(M_{1},M_{2}\right)=\frac{1}{M}\sum_{j=1}^{m}\frac{1}{1+D(M_{1}^{j},M_{2}^{j})} \end{array}$$



**Definition 13** (Entropy of Cell Semantics)**.** 
*Information entropy was proposed by C. E. Shannon to measure the uncertainty of various possible events of the information source [53]. Its value is the statistical average of the event uncertainty (the decrement function of the probability, $f\left(p_{i}\right)=-\log_{2}p_{i}$, computed as in (Equation 6). We introduce entropy as an evaluation criterion to measure the degree of cell set obfuscation (i.e., the attacker’s uncertainty in identifying the precise location) in the anonymity group. The entropy of cell semantics is a metric used to measure the uncertainty of an attacker using regional semantic features to infer the user’s location. It is described using the symbol H. The higher the entropy value, the higher the uncertainty. Theoretically, the maximum entropy value $\frac{1}{k}$ can be reached when the semantic features of cells in the anonymous cell set are identical. The entropy of cell semantics is calculated using the probability of the distribution of the semantic similarity of the cells.*

(6)
$$\begin{array}{c} H=-\sum_{i=1}^{k}p_{i}\log_{2}p_{i} \end{array}$$


(7)
$$\begin{array}{c} p_{i}=\frac{sim(M_{u},M_{i})}{\sum_{i=1}^{k-1}sim\left(M_{u},M_{i}\right)} \end{array}$$

*where $M_{u}$ denotes the cell semantic feature matrix of the user’s cell. $M_{i}$ denotes the cell semantic feature matrix of the assisting user’s cell participating in the anonymous group.*


### 3.4. Ideas and Algorithms of Our Scheme

The core idea of our scheme includes four aspects. First, we use a privacy location instead of a precise location to send check-ins and queries to the AS to reduce the precision with which the AS is informed of the user’s location. Second, we use GL-Tree to optimize the storage and spatial retrieval of user check-ins and queries. Third, we define the cell semantic matrix to inscribe the semantic features of location intervals and select cells similar to the user’s cell to construct the *k*-anonymous cell set. Fourth, we use historical queries to generate fake queries to form a *k*-anonymous query set and implement user identity replacement. The scheme involves two core algorithms: an anonymous cell set generation algorithm and an anonymous query set generation algorithm.

#### 3.4.1. Anonymous Cell Set Generation Algorithm

A user sends a query *Q* to an AS using privacy location *g*. The AS receives the user’s query *Q* and uses this algorithm based on the privacy requirement *p* to generate an anonymous cell set. Algorithm 1 shows the pseudo-code of the algorithm.
**Algorithm 1** Anonymous Cell Set Generation Algorithm**Input:** gltree, a GL-Tree that records user’s check-ins and historical queries; Q, a user’s query**Output:** C, an anonymous cell set;1:$g,g_{max},g_{min},k\leftarrow \;Q$2:$g_{1}\leftarrow g.subString\left(0,g_{max}\right)$3:$g_{2}\leftarrow g.subString\left(0,g_{min}\right)$4:$U\leftarrow getcell(g_{1},\mathrm{gltree})$5:$E\leftarrow getcell(g_{2},\mathrm{gltree})$6:$\mathrm{CS}=\left\{C_{1},C_{2},\ldots ,C_{n}\right\}\leftarrow getSubCells\left(E,g_{max}\right)$7:**for each** 
$C_{i}\;\mathrm{in}\;\mathrm{CS}$
 **do**8:  **if** ($|C_{i}|\leq 0$) **then**9:    $\mathrm{CS}\leftarrow \mathrm{CS}-C_{i}$10:  **else**11:    calculate vectors: ${\mathrm{CH}}_{i}$,${\mathrm{CQ}}_{i}$,${\mathrm{CT}}_{i}$12:    $M_{i}\leftarrow \left[{\mathrm{CH}}_{i},{\mathrm{CQ}}_{i},{\mathrm{CT}}_{i}\right]$13:  **end if**14:**end for**15:sort all cells in CS by the $|C_{i}|$16:**if** (|CS|≥2×k) **then**17:  $\mathrm{CS}=\left\{U,C_{1},C_{2},\ldots ,C_{2k-1}\right\}\leftarrow getCandidateCells\left(\mathrm{CS}\right)$18:**end if**19:**for each** 
$C_{i}\;\mathrm{in}\;\mathrm{CS}$
 **do**20:  calculate $sim(U,C_{i})$21:**end for**22:**if** (|CS|≤k) **then**23:  C←CS24:**else**25:  **for** j←1 to min{k,10}
**do**26:    ${\mathrm{AS}}_{j}=\left\{U,C_{1},C_{2},\ldots ,C_{k-1}\right\}\leftarrow randomSelectCandidateCells\left(\mathrm{CS}\right)$27:    calculate *H*28:  **end for**29:  $C\leftarrow arg^{H_{max}}\left\{{\mathrm{AS}}_{j}\right\}$30:**end if**31:shuffle(C)32:**return:** 
C

The execution process of the Algorithm 1 is described as follows:In line 1, after receiving a user’s query ***Q***, the AS first extracts *g*, $g_{max}$, $g_{min}$, and *k* from ***Q***.In lines 2–5, according to the *p* set by the user, substrings $g_{1}$ and $g_{2}$ are intercepted from *g* and $g_{1}$ and $g_{2}$ are used to retrieve the gltree. The data item nodes corresponding to them in the tree are found, and two cell objects U and E are created corresponding to them, where U and E correspond to the minimum and maximum hidden cells initially defined by the user. U is also the user’s current cell.In lines 6–14, the algorithm finds all data item nodes corresponding to all subgrids of the same size as U in E and a candidate set of subcells corresponding to these data items, CS. The algorithm removes the subcells with 0 check-ins for the current period from CS and calculates ${\mathrm{CH}}_{i}$, ${\mathrm{CQ}}_{i}$, ${\mathrm{CT}}_{i}$ using (Equation 1), (Equation 3), and (Equation 4), respectively, then constructs the cell semantic feature matrix $M_{i}$ of the remaining subcells.In lines 15–18, the algorithm sorts all subcells in the CS based on the number of check-ins in the current period. The algorithm checks whether the number of subcells in the CS is 2k or more. If yes, the algorithm filters out the 2k−1 subcells whose current check-in counts are closest to U and uses them as the initial set of candidate cells along with U.In lines 19–21, the algorithm calculates the semantic similarity of each subcell in the CS and U using the semantic feature matrix.In lines 22–30, the algorithm determines whether there are *k* subcells or less in the CS. If yes, the CS is selected directly. Otherwise, k−1 subcells are chosen randomly from the CS with similarity as the selection weight. Subcells with high similarity also have a high chance of being selected. The selected subcells form an anonymous cell group with U. The selection is randomized for *j* rounds, generating *j* different sets of cells, where *j* is the smaller of *k* and 10. The similarity entropy *H* is evaluated for each group. The group with the maximum entropy is the final selection.In lines 31–32, before outputting the result, the order of the cells in the result set is randomly changed to avoid the attacker speculating U.

#### 3.4.2. Anonymous Query Set Generation Algorithm

After the AS completes the anonymous cell set generation algorithm, it uses this algorithm to create a *k*-anonymous query set. *K* queries are generated by the AS with the Geohash code of each cell as the privacy location *g* in the anonymous cell set. Algorithm 2 shows the pseudo-code of the algorithm.
**Algorithm 2** Anonymous Query Set Generation Algorithm**Input:** gltree, a GL-Tree that records user’s check-ins and historical queries; Q, a user’s query; C, an anonymous cell set output by Algorithm 1;**Output:** Qs, an anonymous query set;1:Qs←Ø2:**while** 
|Qs|<k
 **do**3:  **for each** $C_{i}\;\mathrm{in}\;C$ **do**4:    **if** ($C_{i}.g\neq Q.g$) **then**5:      $q\leftarrow selectHistoryQuery(C_{i},Q,\mathrm{gltree})$6:      **while** *q* is null **do**7:        $C\leftarrow getAdjacentCell(C_{i})$8:        q←selectHistoryQuery(C,Q,gltree)9:      **end while**10:    **else**11:      q←Q12:    **end if**13:    bigint←getRandomBigInteger()14:    $str\leftarrow q.uid+C_{i}.g+Q.t+q.c+bigint$15:    pid←Hash256(str)16:    writeMapTable(q.uid,pid)17:    $Q^{\prime }\leftarrow InitFakeQuery\left(pid,C_{i}.g,Q.t,q.c\right)$18:    $\mathrm{Qs}\leftarrow \mathrm{Qs}⋃Q^{\prime }$19:  **end for**20:**end while**21:**return:** 
Qs

The execution process of the Algorithm 2 is described as follows:In line 1, an empty anonymous query set Qs is initialized.In lines 2–20, if the number of queries in Qs is less than *k*, the algorithm loop generates an anonymous query $Q^{\prime }$ from each cell and adds $Q^{\prime }$ to Qs.In lines 3–12, in addition to the user’s query, the remaining k−1 dummy queries in the anonymous query set are randomly selected from the historical queries in the corresponding cell. If there are no historical queries in the cell, the algorithm selects a query from the historical queries of adjacent cells.In lines 13–17, to further increase the difficulty of attacking the LSP’s side, the algorithm uses the Hash256 algorithm to calculate the Hash value of a string consisting of a combination of q.uid, $C_{i}.g$, q.c, Q.t, and a random large integer. The algorithm creates a dummy query and uses the Hash value as the pid for the dummy query. The algorithm uses an identity mapping table to record the correspondence between q.uid and pid. After getting the query result back from the LSPs, the AS filters the corresponding user query results based on the identity mapping table and sends the results to the user. In line 17, the algorithm constructs a fake query $Q^{\prime }$.In line 21, the algorithm returns Qs.

## 4. Experiments and Analysis

In this section, we evaluated the performance of GLPS by extensive experiments. The detailed experimental results and comments are presented.

### 4.1. Experimental Environment

We used Java language in the Eclipse development environment to complete the code writing and experimental testing. The experiment was run on a Huawei Cloud Elastic Compute Server (HECS) with a four-core (vCPU) Intel(R) Xeo(R) Gold 6278C CPU @2.60 GHz, 8 GiB RAM, a Windows Server 2019 Datacenter 64-bit Operating System, ax64-based processor, and the JavaSE-1.8 environment.

### 4.2. Experimental Dataset

In the experiment, we used a public dataset: GeoLifeTrajectories 1.3 [54]. We randomly selected 1,000,000 records from the Beijing (longitude: 115.8–117.4, latitude: 39.4–40.8) dataset to create a dataset for the experiment. For each experiment, we randomly selected a specified amount of generated check-in history records, and from these, we randomly selected a certain percentage of records to generate history query records. We also set 100 query content types and randomly set one content type for each query. Without loss of generality, we randomly selected a location coordinate within Beijing to simulate a user check in or query.

### 4.3. GLPS Performance Test Experiments

#### 4.3.1. Experimental Method

To test the impact of anonymity *k* and the amount of data on the algorithm’s performance, we randomly selected 100,000, 200,000, 300,000, 400,000, and 500,000 check-in history records from the experimental dataset to simulate the amount of data. We divided the records into 24 periods based on the hour and randomly selected 20% of them to create queries. The location coordinates were randomly encoded into four- to eight-character Geohash codes to simulate different minimum hidden cell requirements. Users send queries by randomly selecting a location coordinate from the location dataset. The maximum and minimum cell set by the user’s query privacy *p* corresponded to Geohash code lengths of four and six, respectively. Experiments were conducted for the same query, varying the value of *k*. We executed the experiment 100 times and took the average of the experimental results for analysis.

#### 4.3.2. Experimental Results

Table 1 and Figure 3 and Figure 4 show the effect of anonymity *k* on the algorithm runtime and the area of the hidden region for different amounts of data. Table 2 and Figure 5 and Figure 6 show the effect of anonymity *k* on the entropy of cells’ user distributions, the entropy of cell semantics, and the anonymous success rate for different amounts of data.

As can be seen from the tables and the figures, for the same *k*, the variation in the amount of data significantly affects the algorithm runtime, while it has a lower effect on the area of the hidden region, the entropy of a cell’s user distribution, and the entropy of cell semantics. With the same amount of data, the algorithm runtime does not change much as *k* increases, while the area of the hidden region, the entropy of a cell’s user distribution, and the entropy of cell semantics all increase significantly. Moreover, we can also see from Table 2 that the anonymous success rate of this scheme can always be maintained at 100% for any *k* and the total amount of data used for the experiment.

Figure 7 shows the entropy of the cell’s user distribution and the cell semantic entropy for a total of 200,000 data. As can be seen from the figure, the cell semantic entropy is larger than the cell’s user distribution entropy under different *k* values.

#### 4.3.3. Results Analysis

GLPS uses GL-Tree to store user check-ins and queries and retrieve information based on the Geohash code. The time complexity of this process is a linear equation of the depth *n* of the GL-Tree tree, denoted as O(n) and independent of the total amount of data stored on the GL-Tree. For a detailed analysis, see [52].

The anonymous cell set generation algorithm locates the data item node corresponding to the user’s maximum hidden cell in the GL-Tree based on *g* and *p* in ***Q***. It branches along this node to retrieve the minimum hidden cell and other equal-sized cells and jointly constructs an anonymous cell set. The time consumption of this retrieval process does not change significantly depending on the amount of data. However, when the amount of data increases, the amount of user check-ins and historical queries in each cell also increases, so the time consumed by the algorithm to compute the feature vectors of each cell and its similarity to the user cell also increases.

The algorithm determines the number of cells in an anonymous cell set based on *k* and selects the final *k*-anonymous cell set based on the maximum cell similarity entropy. The time consumption of this process is affected by the increase in *k*, but it is much smaller than the time consumption of cell feature vector computation. Therefore, the experimental data in Table 1 and the graphical representation in Figure 3 and Figure 4 show that, for the same *k*, an increase in the amount of data leads to a significant increase in the runtime, whereas, for the same amount of data, an increase in *k* affects the increase in the runtime to a lesser extent.

The design aim of GLPS is to hide the user’s location into *k* different but similar cells as much as possible so that an increase in *k* leads to an increase in the area of the final hidden region.

GLPS first selects other cells close to the user’s cell according to the number of users in the current period to form a candidate cell set. On this basis, the algorithm filters the final *k*-anonymous cell set based on the cells’ semantic similarity and maximum entropy. So, the cells in the *k*-anonymous set are more similar in terms of the user’s distribution and semantics, and the semantic entropy of the cells is higher than the entropy of the cell’s user distribution. As *k* increases, the uncertainty of the user’s cell increases, thus making it more difficult for an attacker to determine its location.

GLPS constructs a *k*-anonymous cell set using cells with online users, then forms a *k*-anonymous query set by generating pseudo-queries by selecting historical queries from selected cells or their neighboring cells. When the historical query data and checked-in users in the dataset are distributed in more than *k* cells, the algorithm always generates the final *k*-anonymous query set. The anonymous success rate is always 100%.

This selection strategy can avoid the irrationality of location distributions caused by the random selection strategy (e.g., for the current cycle, the selected location is an off-the-beaten-path location), and it can also ensure that the selected locations are distributed in as many cells as possible, avoiding a centralized distribution. Therefore, the algorithm can reduce the risk of background knowledge attacks, distribution density attacks, and homogeneity attacks.

### 4.4. Comparative Experiments with Different Schemes

To verify whether GLPS is superior in terms of performance, we chose Casper, GCasper, and DLS, which have high correlations, for comparison experiments.

#### 4.4.1. Experimental Method

We conducted comparative experiments based on the same dataset and geographic area grid division criteria. We randomly used 500,000 history records and created 50,000 online users, with 20% of the online users submitting queries. In the experiment, the minimum and maximum hidden cells of users were limited, which correspond to a 7-bit Geohash code interval (i.e., 0.1525×0.1529 km${}^{2}$) and a 5-bit Geohash code interval (i.e., 4.9×4.9 km${}^{2}$), respectively. By adjusting the anonymity *k*, the above four schemes were tested in terms of four metrics: the area of the hidden region, the anonymous success rate, the entropy of the cell’s user distribution, and the entropy of cell semantics.

#### 4.4.2. Experimental Results

Figure 8 and Figure 9 show the experimental results of the four schemes. Figure 8 shows that the area of the hidden region generated by all schemes increases as *k* increases, but when using GLPS and DLS, they increase faster and more closely. GLPS generates a slightly smaller area of the hidden region than DLS for larger *k* values. Figure 9 shows that as *k* increases, the anonymous success rate of GLPS and DLS is consistently at 100%, while the anonymous success rate of Casper and GCasper tends to decrease. Figure 10 and Figure 11 show that the entropy of the user distribution and the entropy of cell semantics in the anonymous cell set both tend to increase as *k* increases. GLPS always has the largest entropy for the same *k*.

#### 4.4.3. Results Analysis

As *k* increases, the anonymous region needs to contain more cells and users to satisfy the privacy requirement, and the area of the hidden region will increase trend. Since the Casper and GCasper schemes only consider the number of online users in cells, user location aggregation is allowed in the same selected cell. DLS and GLPS spread the user across as many cells as possible so that more cells are selected, and the final hidden area is larger.

The experimentally limited maximum hidden cell affects the anonymous success rate. For Casper and Gcasper, based on the algorithmic idea, when *k* increases, the number of users selected within this maximum hiding area may not satisfy the *k* requirement, resulting in anonymization failure. Therefore, the anonymous success rate will have a decreasing trend. However, for the maximum hidden cells, DLS uses the distribution of users in the history to select candidate subcells, and GLPS searches for as many subcells with online users as possible to construct an anonymous cell set and then selects the history queries among the selected cells to generate an anonymous query set. As a result, there are more optional cells, and the anonymous success rate is always 100%.

When creating an anonymous cell set, DLS selects cells based on the user distribution density only, while GLPS further considers the semantic features of cells from time series based on the distribution density of the current online users and calculates the entropy of the semantic similarity for the final selection. Casper and GCasper select cells based on the number of current users and do not consider the user distribution or cell semantic features. Therefore, the selection results of GLPS and DSL have obvious advantages regarding the entropy value, and GLPS performs better.

## 5. Conclusions

Current, mainly centralized location privacy protection schemes rely on the assumption of absolute security and trustworthiness of a central anonymous server; this assumption is too idealized and usually not applicable. In this paper, to solve this strong dependency problem, we design a location privacy protection scheme, GLPS, that satisfies users’ personalized needs. GLPS not only utilizes Geohash code to hide the exact location of a user but also incorporates the design idea of *k*-anonymity by selecting *k*-cells with a similar distribution of online users and similar cell semantics to hide the user’s actual location, which decreases the probability of an attacker identifying or guessing a user’s location. GLPS has the following advantages:In GLPS, users can use Geohash codes with different lengths to satisfy personalized hidden space requirements, so GLPS has greater flexibility and personalization.GLPS selects fake locations scattered over a wider geographic area, avoiding a centralized and uneven distribution. The selected cells are those with online users in the current period, and all selected cells have similarities in user distribution and cell semantics. Generated fake queries use historic locations and query contents, which makes it more difficult for the attacker to recognize them. The anonymous query set is resistant to background knowledge attacks, region-centric attacks, homogenization attacks, and distribution density attacks.User identities in the anonymous query set are replaced with a pseudo-identity, and different queries use different pseudo-identities, which makes it impossible for an attacker to track user’s trajectories to implement identity association attacks.

In continuous queries, trajectories are more likely to leak users’ private data. Therefore, in the research in this paper, we have thought about the problem of trajectory privacy protection. We believe that the location approximation capability of Geohash encoding can be utilized for segmentation, filtering, classification, and similarity computation of trajectories, but this requires the targeted design of feasible schemes and quantitative computation criteria. Our future work will focus on the application of Geohash encoding for trajectory privacy protection.

## Figures and Tables

**Figure 1 entropy-25-01569-f001:**
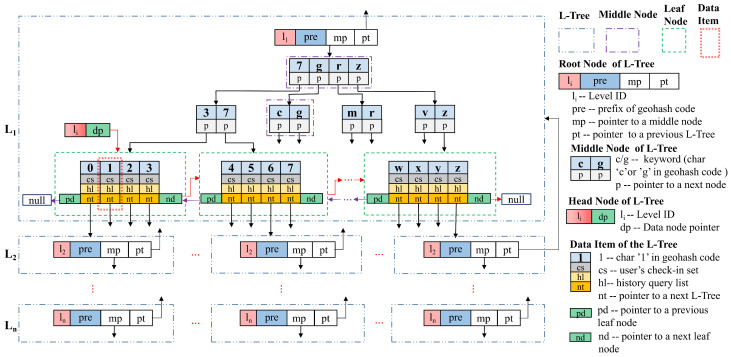
The modified structure of GL-Tree.

**Figure 2 entropy-25-01569-f002:**
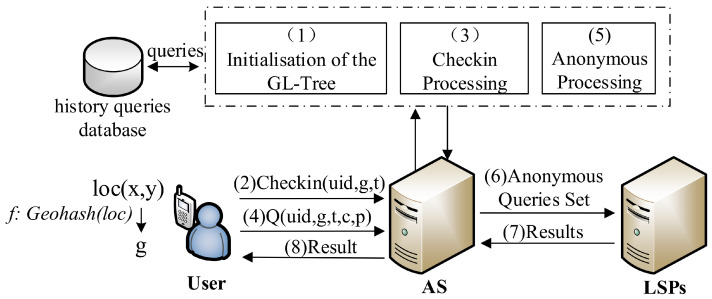
The system architecture.

**Figure 3 entropy-25-01569-f003:**
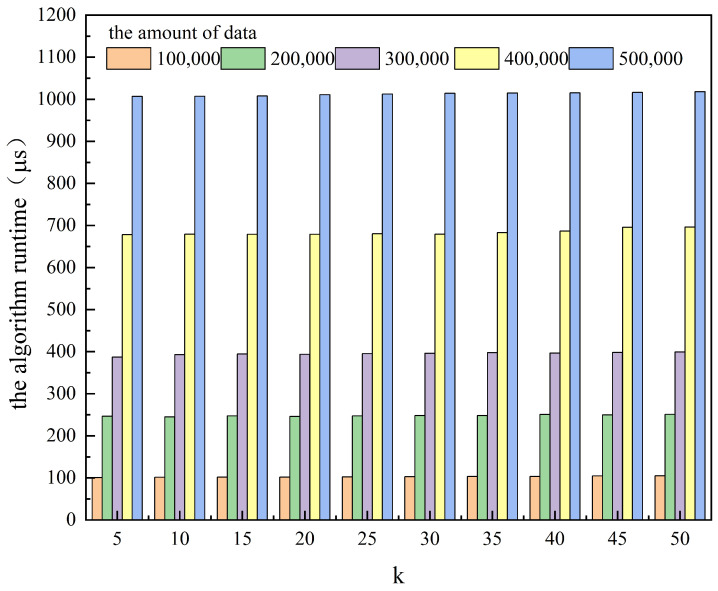
The effect of anonymity *k* on the algorithm runtime for different amounts of data.

**Figure 4 entropy-25-01569-f004:**
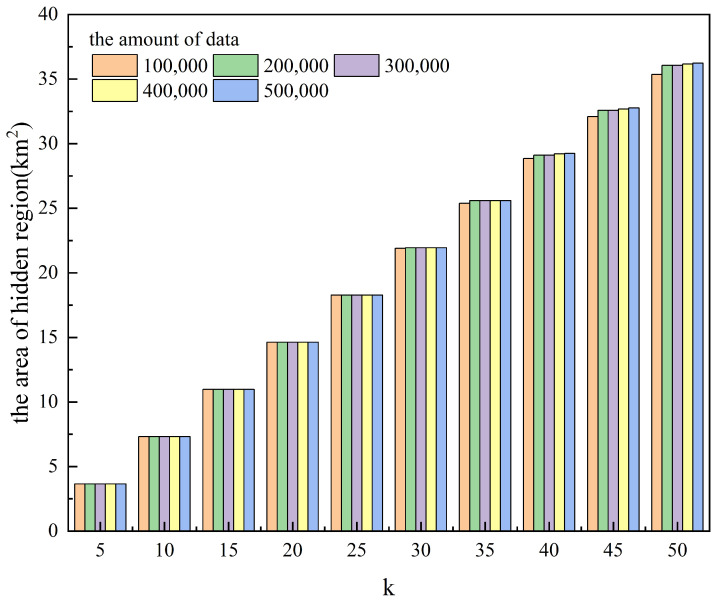
The effect of anonymity *k* on the area of the hidden region for different amounts of data.

**Figure 5 entropy-25-01569-f005:**
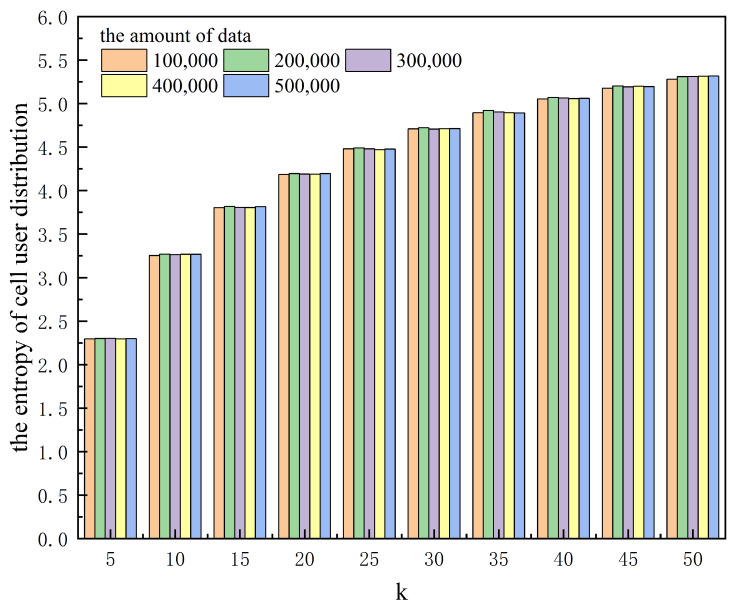
The effect of anonymity *k* on the entropy of the cell’s user distribution for different amounts of data.

**Figure 6 entropy-25-01569-f006:**
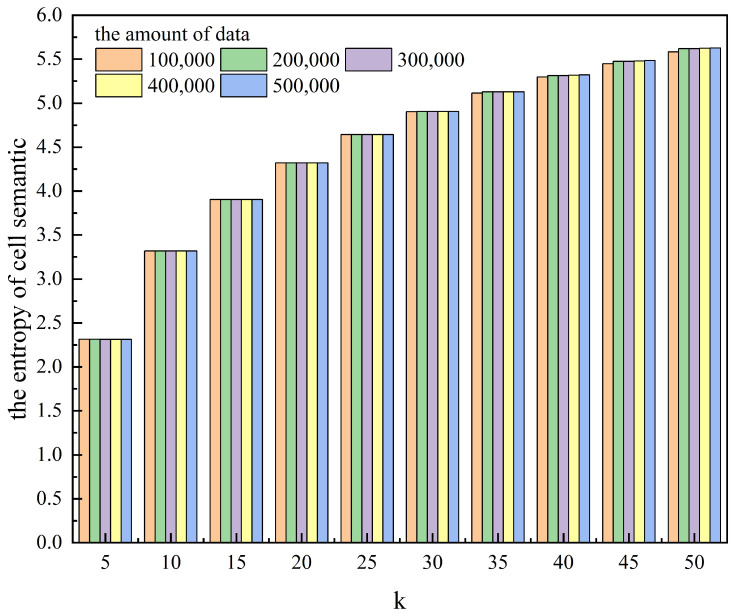
The effect of anonymity *k* on the entropy of cell semantics for different amounts of data.

**Figure 7 entropy-25-01569-f007:**
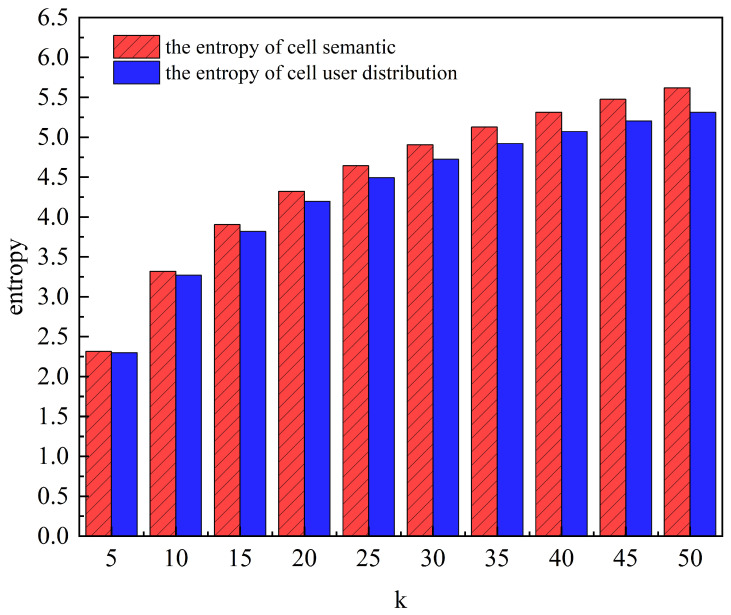
The effect of anonymity *k* on the two kinds of entropy.

**Figure 8 entropy-25-01569-f008:**
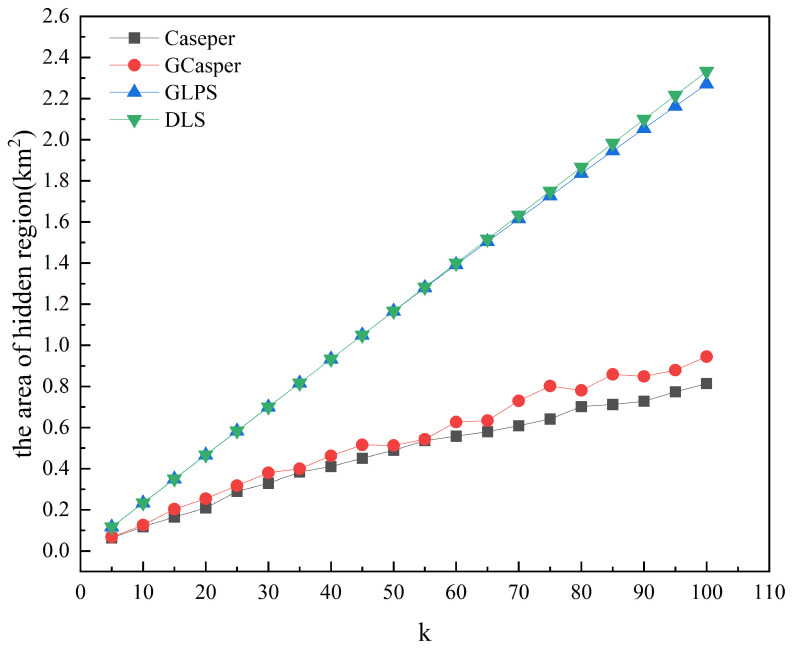
The effect of increasing anonymity *k* on the area of the hidden region.

**Figure 9 entropy-25-01569-f009:**
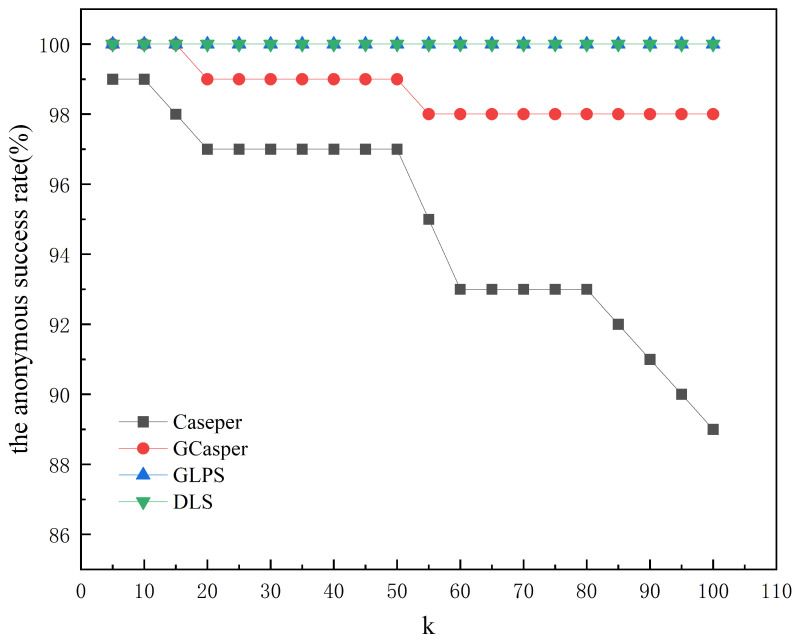
The effect of increasing anonymity *k* on the anonymous success rate.

**Figure 10 entropy-25-01569-f010:**
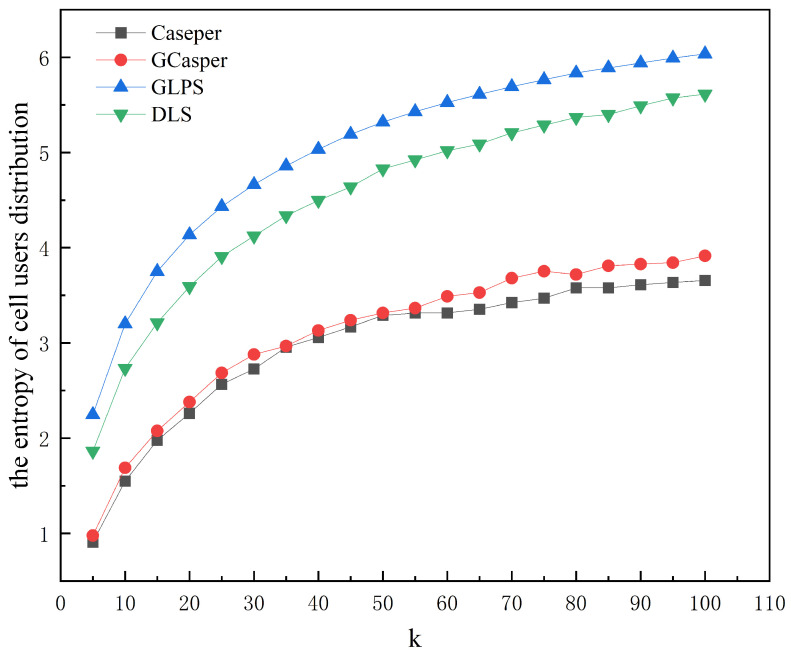
The effect of increasing anonymity *k* on the entropy of the cell’s users distribution.

**Figure 11 entropy-25-01569-f011:**
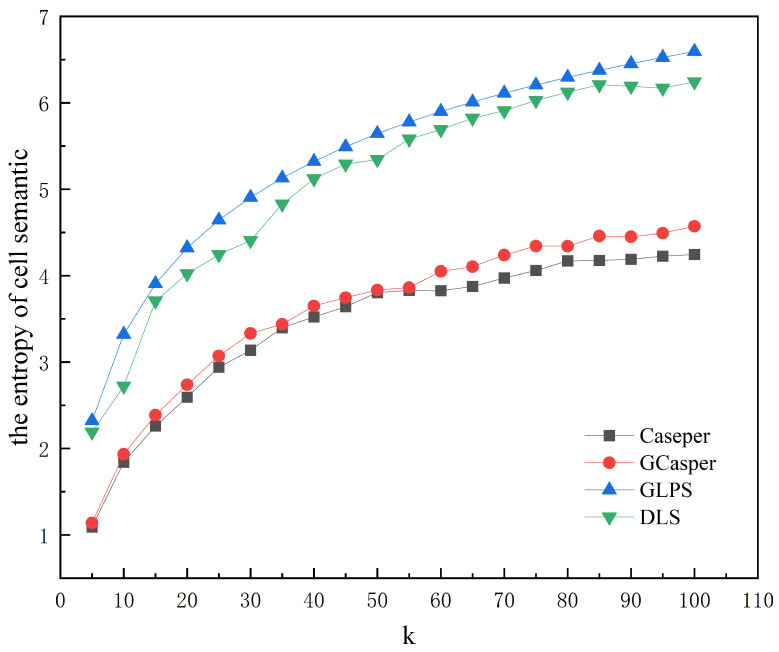
The effect of increasing anonymity *k* on the entropy of cell semantics.

**Table 1 entropy-25-01569-t001:** The effect of anonymity *k* on the algorithm runtime and the area of the hidden region for different amounts of data.

	Amount of Data									
	100,000		200,000		300,000		400,000		500,000	
*k*	T	A	T	A	T	A	T	A	T	A
5	100.53	3.66	246.42	3.66	387.15	3.66	678.18	3.66	1006.79	3.66
10	101.35	7.31	245.11	7.31	393.03	7.31	679.20	7.31	1007.12	7.31
15	101.83	10.97	247.32	10.97	394.40	10.97	679.06	10.97	1008.12	10.97
20	101.82	14.63	246.15	14.63	393.81	14.63	678.95	14.63	1010.90	14.63
25	102.59	18.28	247.41	18.28	395.19	18.28	680.34	18.28	1012.48	18.28
30	102.85	21.90	247.89	21.93	396.34	21.93	679.24	21.93	1014.18	21.93
35	103.32	25.38	248.14	25.59	397.72	25.59	683.15	25.59	1014.84	25.59
40	103.43	28.84	250.76	29.10	396.45	29.10	686.47	29.21	1015.30	29.25
45	104.62	32.09	249.89	32.58	398.21	32.58	695.63	32.69	1016.13	32.76
50	104.94	35.35	251.05	36.05	399.30	36.05	696.24	36.16	1018.06	36.23

T—the algorithm runtime (µs); A—the area of hidden region (km${}^{2}$).

**Table 2 entropy-25-01569-t002:** The effect of anonymity *k* on the entropy of cell’s user distribution, cell semantic entropy, and anonymous success rate for different amounts of data.

	Amount of Data														
	100,000			200,000			300,000			400,000			500,000		
*k*	D	S	R	D	S	R	D	S	R	D	S	R	D	S	R
5	2.29	2.31	100	2.30	2.31	100	2.30	2.31	100	2.29	2.31	100	2.30	2.31	100
10	3.25	3.32	100	3.27	3.32	100	3.26	3.32	100	3.27	3.32	100	3.27	3.31	100
15	3.80	3.90	100	3.82	3.90	100	3.81	3.90	100	3.81	3.90	100	3.82	3.90	100
20	4.19	4.32	100	4.20	4.32	100	4.19	4.32	100	4.19	4.32	100	4.20	4.32	100
25	4.48	4.64	100	4.49	4.64	100	4.48	4.64	100	4.47	4.64	100	4.48	4.64	100
30	4.71	4.90	100	4.72	4.91	100	4.71	4.91	100	4.71	4.91	100	4.71	4.91	100
35	4.89	5.11	100	4.92	5.13	100	4.90	5.13	100	4.90	5.13	100	4.89	5.13	100
40	5.05	5.29	100	5.07	5.31	100	5.06	5.31	100	5.06	5.32	100	5.06	5.32	100
45	5.18	5.45	100	5.20	5.47	100	5.19	5.47	100	5.20	5.48	100	5.20	5.48	100
50	5.28	5.58	100	5.31	5.62	100	5.31	5.62	100	5.31	5.62	100	5.32	5.63	100

D—the entropy of the cell’s user distribution; S—the entropy of cell semantics; R—the anonymous success rate (%).

## Data Availability

Data are contained within the article.

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
