# Peer review of "GLPS: A Geohash-Based Location Privacy Protection Scheme"

_entropy, 2023, doi:10.3390/e25121569_

Round 1
Reviewer 1 Report
Comments and Suggestions for Authors
The paper describes a scheme for Geo-hashing of location queries for privacy protection, using GL-trees.
The paper is well written and organized and the authors have done a good job describing the field, the related concepts and related works. With regards to related works, even though the cited references are relevant, it is notable that many seminal works in this domain are not cited.
With regards to the description of the scheme, although it is well understood and clear, the same description (or more or less the same) is repeated in a few sections. I would make the report more concise by removing unnecessary redundancies of descriptions.
The authors have conducted many experiments to test the efficacy of their scheme. One thing missing, is to present the anonymization success rate for the various k values and the various data-amount trials, as they present it in the comparative study.
Comments on the Quality of English LanguageEnglish is fine, with minor typos and syntax errors here and there, but the paper is easy to read and generally well-written.
Author Response
Dear Reviewer.
Thank you very much for your review of this article. It has been revised based on your comments. The specific revisions are described as follows.
Response to your valuable suggestions:
Suggestion 1:With regards to related works, even though the cited references are relevant, it is notable that many seminal works in this domain are not cited.
Response :
In response to this suggestion, we have reorganized the INTRODUCTION section into three subsections.
1.1 The Typical Location Privacy Protection Technologies
1.2 The Research on Related Location Privacy Protection Solutions
1.3 The Research of Geohash in Location Services and Privacy Protection
The first subsection is a new addition that introduces typical location privacy protection techniques, including k-anonymity, dummy, spatial cloaking, spatial transformation, release suppression, PIR(privacy information retrieval), encryption, differential privacy, etc. The basic ideas of these techniques are supplemented with a large number of applications from classical highly cited literature.
Suggestion 2:With regards to the description of the scheme, although it is well understood and clear, the same description (or more or less the same) is repeated in a few sections. I would make the report more concise by removing unnecessary redundancies of descriptions.
Response :
In response to this suggestion, we merged “3.4 The Idea of Our Scheme ” and “3.5 The Algorithm Design” into a single subsection “3.4 Ideas and Algorithm of Our Scheme”. In the first paragraph before the algorithm, we highlight the four core ideas of the scheme and remove repetitive content. In addition, some of the repetitive text in the experimental section has been deleted.
Suggestion3: The authors have conducted many experiments to test the efficacy of their scheme. One thing missing, is to present the anonymization success rate for the various k values and the various data-amount trials, as they present it in the comparative study.
Response :
In response to this suggestion, we re-targeted our experiments and again determined that this scheme achieves 100% anonymity for different data totals and k cases. We add the results of our experiments in Table 2 and discuss them in a targeted analysis in the Experimental Analysis section.
Suggestion4: English is fine, with minor typos and syntax errors here and there, but the paper is easy to read and generally well-written.
Response :
We carefully examined the paper and found and corrected several grammatical and word spelling errors.
The above is a description of the changes made based on your comments. Thank you again for your hard work.
Kind regards
The author
Reviewer 2 Report
Comments and Suggestions for Authors
This paper discusses location privacy protection systems with centralized architectures assume that anonymous servers are secure and reliable. To solve the problem of relying on the security and reliability of anonymous servers, it proposes a Geohash-based location privacy protection scheme for GLPS instant queries.
It seems interesting to use Geohash encoding technology to convert the user's location coordinates into a string code representing a rectangular geographical area. GLPS uses the code as a private location to send check-ins and queries to the anonymous server and prevent the server from obtaining the user's exact location.
Experimental results and analysis show that GLPS appears to have good performance and privacy protection capability, which solves the dependency on the security and reliability of anonymous servers. According to them it seems to be resistant to attacks related to background knowledge, regional centers, homogenization, distribution density and identity association.
The work seems interesting although nothing is mentioned about the possibility of relying on servers protected with secure elements such as TPM as is done in "P2ISE: Preserving Project Integrity in CI/CD Based on Secure Elements".
This paper discusses location privacy protection systems with centralized architectures assume that anonymous servers are secure and reliable. To solve the problem of relying on the security and reliability of anonymous servers, it proposes a Geohash-based location privacy protection scheme for GLPS instant queries.
It seems interesting to use Geohash encoding technology to convert the user's location coordinates into a string code representing a rectangular geographical area. GLPS uses the code as a private location to send check-ins and queries to the anonymous server and prevent the server from obtaining the user's exact location.
Experimental results and analysis show that GLPS appears to have good performance and privacy protection capability, which solves the dependency on the security and reliability of anonymous servers. According to them it seems to be resistant to attacks related to background knowledge, regional centers, homogenization, distribution density and identity association.
The work seems interesting although nothing is mentioned about the possibility of relying on servers protected with secure elements such as TPM as is done in "P2ISE: Preserving Project Integrity in CI/CD Based on Secure Elements".
Author Response
Dear Reviewer.
Thank you very much for your review of this article. It has been revised based on your comments. The specific revisions are described as follows.
Response to your valuable suggestions:
Suggestion 1:The work seems interesting although nothing is mentioned about the possibility of relying on servers protected with secure elements such as TPM as is done in "P2ISE: Preserving Project Integrity in CI/CD Based on Secure Elements".
Response :
In response to this suggestion, we describe in the first paragraph of Section 1.1 the three dimensions of security research for the application service lifecycle, i.e., Deployment Security, Operational Security, and Disaster Recovery, and illustrate the different points of security considerations for location privacy protection. In the deployment security, we cite "P2ISE: Preserving Project Integrity in CI/CD Based on Secure Elements".
The above is a description of the changes made based on your comments. Thank you again for your hard work.
Kind regards
The author
Reviewer 3 Report
Comments and Suggestions for Authors
To avoid the risks associated with using untrusted servers for Location Based Serrvices, the authors propose a Geohash-based location privacy protection scheme, GLPS, for geogrpahical position queries and serives. Using Geohash encoding technology, GLPS transforms the user's latitude and longitude into a string code that represents a rectangle area on the map. To ensure that the anonymous server does not learn the precise location of the user, GLPS uses the code as the privacy location while transmitting check-ins and queries to the server. For interval localization and data storage, the GLPS use the GL-Tree to keep track of user check-ins and queries, retrieving the cell where the user is based on the Geohash code of the user's location.
The paper is well written and addresses an important problem. My worry is the overlap with a previous paper by the same authors:
GL-Tree: A Hierarchical Tree Structure for Efficient Retrieval of Massive Geographic Locations
Sensors 2023, 23(4), 2245; https://doi.org/10.3390/s23042245
I am not certain that the present submission is so enhanced with respect to the previous paper, which presents the GL-tree idea and the relevant management algorithms.
Author Response
Dear Reviewer.
Thank you very much for your review of this article.In response to your question, we answer the following:
Response :
(1) GL-Tree: A Hierarchical Tree Structure for Efficient Retrieval of Massive Geographic Locations
In this paper, we mainly focus on the efficient storage and retrieval needs of massive locations. We described the related tree structure research and applications. Then, we describe in detail the structure, physical significance, illustrative examples of creation and retrieval, and corresponding algorithms of GL-Tree, which is designed by combining the principle of Geohash coding. We discuss the algorithmic complexity of GL-Tree and its possible applications for location privacy protection. In the experimental section, we experimented and analyzed the effectiveness of GL-Tree in comparison with B+-tree and RTree at three levels: tree creation, position insertion, and position retrieval.
(2) GLPS:A Geohash-based Location Privacy Protection Scheme
In this paper, we focus on the problem of how to solve the centralized location privacy protection scheme's reliance on anonymity servers' high trustworthiness and security by proposing a new location privacy protection scheme, GLPS. We described the system architecture, adversary assumptions, formal description, algorithmic ideas, pseudo-code, algorithmic analysis, and experiments. GL-Tree is just the data structure used by the anonymous server to store users' check-ins and queries in the GLPS. To make GL-Tree suitable for this scheme, we adjusted its data item part by adding 'cs' and 'hl', the specific Java code implementation of 'cs' uses TreeSet for storing the check-in information, and 'hl' uses ArrayList for storing the query requests.
Therefore, this paper is a new research based on the research results of the previous paper, which is quite different from research objectives, research content, experimental content, and comparison subjects.
Thank you again for your hard work.
Kind regards
The author
Round 2
Reviewer 3 Report
Comments and Suggestions for Authors
The authors demonstrated clearly, in the new version of the paper, their contribution compared to their previous work, which was my concern.